# Learning Behaviors
# through Physics-driven Latent Imagination

**Antoine Richard**
Georgia Institute of Technology GA, USA
`antoine.richard@gatech.edu`

**Stéphanie Aravecchia**
IRL 2958 GT-CNRS, Metz, France

**Matthieu Geist**
Google Research, Brain Team

**Cédric Pradalier**
IRL 2958 GT-CNRS, Metz, France

**Abstract:** Model-based reinforcement learning (MBRL) consists in learning a so-called world model, a representation of the environment through interactions with it, then use it to train an agent. This approach is particularly interesting in the context of field robotics, as it alleviates the need to train online, and reduces the risks inherent to directly training agents on real robots. Generally, in such approaches, the world encompasses both the part related to the robot itself and the rest of the environment. We argue that decoupling the environment representation (for example, images or laser scans) from the dynamics of the physical system (that is, the robot and its physical state) can increase the flexibility of world models and open doors to greater robustness. In this paper, we apply this concept to a strong latent-agent, Dreamer. We then showcase the increased flexibility by transferring the environment part of the world model from one robot (a boat) to another (a rover), simply by adapting the physical model in the imagination. We additionally demonstrate the robustness of our method through real-world experiments on a boat.

**Keywords:** Model-Based Reinforcement Learning, Field Robotics, Latent Models

## 1 Introduction

Early forms of controllers, such as classic optimal control or dynamic programming, make use of a supposedly known model of the environment. On the learning side, Reinforcement Learning (RL) can also learn a model and use it to train an agent. It is only during the last decade that Model-Based RL (MBRL) algorithms capable of controlling systems using high dimension inputs, such as raw images, were developed. These advances allowed their use on robots for solving complex tasks, on which traditional methods failed [1, 2]. In robotics, it is of paramount importance to learn quickly, as acquiring samples is not only ludicrously time-consuming, but can also be dangerous for both the robot and the operator overseeing the agent. Hence, due to their high sample efficiency and their potential resort to offline learning [3, 4], world models make for a compelling choice in robotics applications.

Yet, because most of the agents are meant to be used on classic benchmarks such as Atari [5], OpenAI Gym [6] or the DeepMind control suite [7], they process either images or low proprioceptive variables (for example, position or speed), but rarely both. In robotics, most of the works that used MBRL for solving high-dimensional tasks discard the proprioceptive variables [1, 2, 8], even though they could have been acquired fairly easily. Other works, such as [9, 10], make use of both, but embed them as a single model. Unfortunately, with neural networks there is no way of accessing the dynamics of the system directly integrated inside the environment because they make for a single black-box model. All in all, this means that the world model learned by an agent will be unique to this system.

By contrast, we argue that a world-model should be made of two components: an environment, which encompasses everything that the agent can sense using laser-scanners or cameras, and the dynamics, i.e. the set of physical equations and variables that rule the movement of the mobile robot in the world. To do so, within the paradigm of Markov Decision Processes (MDPs), we decompose the total state of the system in two parts, one related to the environment and one to the dynamics of the

5th Conference on Robot Learning (CoRL 2021), London, UK.

robot, $S^{\text{tot}} = (S_t^{\text{env}}, S_t^{\text{dyn}})$, and we assume that the transition from state to state can be decomposed as a change of the dynamics in response to the applied action, and a change of the environment in response to the new dynamics, $P(S_{t+1}^{\text{tot}}|S_t^{\text{tot}}, a_t) = P(S_{t+1}^{\text{dyn}}|S_t^{\text{dyn}}, a_t)P(S_{t+1}^{\text{env}}|S_t^{\text{env}}, S_t^{\text{dyn}})$. A concrete example of why such representations are powerful is to think of car manufacturers. All manufacturers make different cars with different physical properties, yet those cars all share the same environment: the road. Hence, a method like ours could allow the cars to share their environment while having their own dynamics, speeding up the training process and potentially allowing the distributed training of heterogeneous agents. Furthermore, it makes sense in robotics applications to use proprioceptive information, as most real robotic tasks involve low level physical constraints coupled to high level perception constraints. In the end, MBRL is a very promising solution to bring more autonomy to real-robots, yet we think it would be better suited to mobile robotics application if it was to natively account for the robots' dynamic.

In this paper, we modify one of the strongest latent-model agent, Dreamer [11], and integrate the dynamics as an independent part of the world model. As such, our world model will feature two states: an environment state that uses low dimensions proprioceptive variables to transition from one state to another ($P(S_{t+1}^{\text{env}}|S_t^{\text{env}}, S_{t+1}^{\text{dyn}})$), and a physical state, that uses the action sent by the agent to transition from one state to another ($P(S_{t+1}^{\text{dyn}}|S_t^{\text{dyn}}, a_t)$). While it might seem fairly similar to the original world model with an observation of the proprioceptive input in addition to the images, it is not. Decoupling the dynamics from the environment allows for interesting manipulations within the imagination process used to learn the actor. Because the dynamics is no longer attached to the environment, we can now swap the dynamic model of a robot for the dynamic model of another robot as long as their proprioceptive states are of the same size.. Moreover, our method ensures that the imagination roll-out have environment-states that are consistent with the physical-states. Eventually, we can use an analytic approximation of the robot dynamic model to alleviate the need to rely on a learned model.

Our contributions can be summarized as follows: **(1)** we propose a latent-state representation that decouples the environment from the dynamics to better fit mobile robots; **(2)** we test this method in simulation and on a real robot and show that it robustifies the agents compared to the baseline; and **(3)** we demonstrate that this method can also be used to transfer an environment from one robot to another in a zero-shot setup. All the code to reproduce those experiments is open-source and available on github (Appx. D.1).

## 2   Related Work

**Model-Based Reinforcement Learning.**   The earliest form of Model-Based Reinforcement Learning (MBRL) is probably optimal control. In optimal control, the model of the system is typically known, and an optimal controller is computed using methods such as LQR. Until recently MBRL algorithms such as Pilco [12], among others [13], were designed to process and model low dimensional inputs. These methods were applied to robotic applications [14, 15] with impressive results, but they were limited and struggled to control systems using high dimension sensory inputs such as images. However, with the advent of deep-neural-network, MBRL was applied successfully to high dimensions inputs, as shown in the comprehensive survey of [16]. As of today, MBRL performances are nearing if not exceeding the performances of model-free agents [11, 17]. The main advantage of MBRL relies on its higher sample efficiency when compared to the model-free agents. Methods based on Variational Auto-Encoders (VAE) [18], such as latent state methods [19, 11, 17], can learn robust policies thanks to the VAE stochastic states. Unfortunately, as these methods started processing higher dimension inputs, they also started to disregard proprioceptive information as it made solving the task easier.

**Dreaming applied to real-world.**   Latent-state methods have shown impressive results when applied on real robots. [8] applies Dreamer on a 1/10th racecar with a LIDAR and compares it to other model-free methods in simulation. They show that, on their task, Dreamer performs better than other model free RL algorithms, such as DDPG [20] or PPO [21]. They also demonstrate that Dreamer transfers well from simulation to real robots, in well controlled environments. [2] also applies Dreamer on an Unmanned Surface Vessel (USV) with a LIDAR and trains it in simulation on a shore following task. They then deploy it on the real robot, without retraining it, and evaluate the robustness of the approach by both changing the dynamics of the real systems and evaluating it in different weather conditions. Both of these works also use Dreamer as their backbone, but unlike

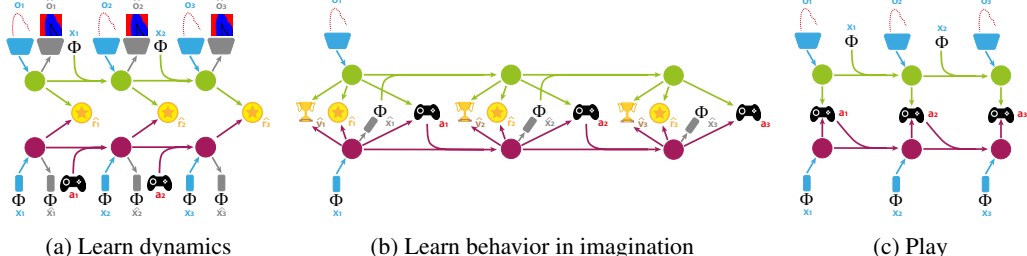

|     (a) Learn dynamics     |     (b) Learn behavior in imagination     |     (c) Play     |

Figure 1: The three concepts of the proposed extension of Dreamer: (a) Using its replay buffer, the agent learns to encode environment high-dimension sensory observation and proprioceptive input into a compact latent environment state (●). Additionally, using a similar process, the agent learns to encode proprioceptive inputs and actions into a latent physical state (●). Both of these states are then used to estimate the reward (★). (b) Using both learned latent spaces, the agent predicts state values (🏆) and actions (🎮) as in the original Dreamer. (c) The agent observes its environment and predicts the best action. More details about the algorithms used here can be found in Appx. C.

ours, they completely disregard the physical state of the system. [1] shows an application of latent models to a real world system using cameras. In [9], they rely on latent state model and integrate both the physical state of the system and high level distance sensors to move their robot around. However, they still consider the systems dynamics to be part of the whole model. A work that is closer to ours is [22], in which they learn from images the Lagrangian dynamic of a system, and then use it to reconstruct the observed images. In its current state, [22] is not applicable to real robots: the examples are simplistic, and make strong approximations about the dynamic model. [23] also shares similarities with the work presented here: they use Graph Neural Networks to perform system identification as in [24], and then apply a Model Predictive Controller and RL on the predicted physical states. The main difference with our work, is that we don't only have a dynamic module but also a perception module. In our method, the outputs of the dynamic module are used to predict the next perception state, and both of their latent states are then used by the actor to decide on which action to apply next.

## 3 Method

We now show how our model differs from the original implementation of Dreamer[11], which we started from. As in Dreamer, we solve a Partially Observable Markov Decision Process (POMDP), with discrete time steps $t$, for which we have continuous actions $a_t$, high dimension sensory inputs $o_t$, low dimension proprioceptive inputs $x_t$, and rewards $r_t$ generated by the environment.

Similarly to Dreamer, our version is articulated around three main concepts: the learning of the world model from previous experiences (Fig. 1a), the learning of the behavior inside the imagination process (Fig. 1b), and finally the application of the policy using observations from the environment to collect new samples (Fig. 1c).

Unlike the original Dreamer, our agent does not rely on a single latent state. Instead, we consider that this single state can be decomposed into two states $S_t^{\text{tot}} = (S_t^{\text{env}}, S_t^{\text{dyn}})$, where $P(S_{t+1}^{\text{tot}}|S_t^{\text{tot}}, a_t) = P(S_{t+1}^{\text{dyn}}|S_t^{\text{dyn}}, a_t)P(S_{t+1}^{\text{env}}|S_t^{\text{env}}, S_t^{\text{dyn}})$. In practice, we consider that we can learn a function $q_\phi(x_t|S_t^{\text{dyn}})$, this means that we can further decompose $P(S_{t+1}^{\text{tot}}|S_t^{\text{tot}}, a_t) = P(S_{t+1}^{\text{dyn}}|S_t^{\text{dyn}}, a_t)P(S_{t+1}^{\text{env}}|S_t^{\text{env}}, x_t)$ as we will be able to estimate $x_t$ from $S_t^{\text{dyn}}$ within the imagination. In [11, 17], the latent dynamic model is composed of three modules: a representation module $p_\eta(s_t \mid s_{t-1}, a_{t-1}, o_t)$, a transition module $q_\eta(s_t \mid s_{t-1}, a_{t-1})$, and a reward module $q_\eta(r_t \mid s_t)$. With this setup, both the representation and transition modules embed the dynamics of the system and the understanding of the environment. This limits the capacities of the world model. Hence, we propose to separate the dynamics of the system from the environment. To do so, we change the latent dynamic model of Dreamer to include two states: $S_t^{env}$ the environmental state, and $S_t^{dyn}$ the physical state. We then add two extra modules and slightly modify the inputs of the original latent dynamic model.

The new latent dynamic model is

$$
\begin{aligned}
\text{Dynamics representation module:} \quad & p_\phi(S_t^{dyn} \mid S_{t\text{-}1}^{dyn}, a_{t\text{-}1}, x_t) \\[4pt]
\text{Dynamics transition module:} \quad & q_\phi(S_t^{dyn} \mid S_{t\text{-}1}^{dyn}, a_{t\text{-}1}) \\[4pt]
\text{Environment representation module:} \quad & p_\eta(S_t^{env} \mid S_{t\text{-}1}^{env}, x_{t\text{-}1}, o_t) \\[4pt]
\text{Environment transition module:} \quad & q_\eta(S_t^{env} \mid S_{t\text{-}1}^{env}, x_{t\text{-}1}) \\[4pt]
\text{Reward module:} \quad & q_\eta(r_t \mid S_t^{env}, S_t^{dyn}).
\end{aligned}
\tag{1}
$$

**Learning dynamics & Reconstructing Observations.** To learn both the dynamics and the environment, we rely on Recurrent State Space Models (RSSM) [19]. RSSMs can be seen as non-linear Bayesian filter [25], with an observation step and a prediction step. During the observation step, the RSSM is given its previous state and an observation, and outputs a compact latent state. During the prediction step, the RSSM is given its previous state and command, and outputs a compact latent state. This prediction step of the RSSM is the transition function, and the combination of a prediction and an observation is the representation module.

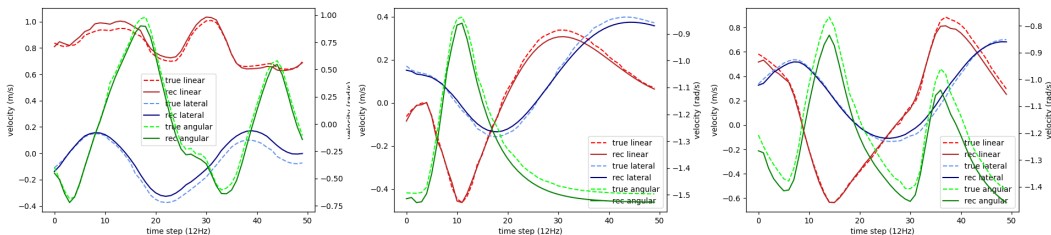

Figure 2: RSSM reconstruction of proprioceptive variables, 5 observations followed by 45 predictions.

We consider two RSSMs: one learns the dynamics of the system and the other learns the environment.

(●) The RSSM that learns the dynamics takes as input both the raw proprioceptive variables and the actions. To teach the network to embed within its latent state relevant information, we also train a decoder to reconstruct the proprioceptive variables associated with that state. Fig. 2 shows examples of reconstructed states.

$$
\begin{aligned}
\text{Dynamics observation module:} \quad & q_\phi(S_t^{dyn} \mid S_{t\text{-}1}^{dyn}, x_t) \\[4pt]
\text{Dynamics transition module:} \quad & q_\phi(S_t^{dyn} \mid S_{t\text{-}1}^{dyn}, a_{t\text{-}1}) \\[4pt]
\text{Dynamic reconstruction module:} \quad & q_\phi(\hat{x}_t \mid S_t^{dyn}).
\end{aligned}
\tag{2}
$$

(●) The RSSM that learns the environment takes as input a high dimension sensory observation and the latent dynamic state. We then train another network to reconstruct from its latent state the inputs or a projection of it.

$$
\begin{aligned}
\text{Environment observation module:} \quad & q_\eta(S_t^{env} \mid S_{t\text{-}1}^{env}, o_t) \\[4pt]
\text{Environment transition module:} \quad & q_\eta(S_t^{env} \mid S_{t\text{-}1}^{env}, x_{t\text{-}1}) \\[4pt]
\text{Environment reconstruction module:} \quad & q_\eta(\hat{o}_t \mid S_t^{env}).
\end{aligned}
\tag{3}
$$

Using these two latent states, $S^{env}$ and $S^{dyn}$, we also learn to predict the reward that will be given by the environment $q_\eta(r_t \mid S_t^{env}, S_t^{dyn})$. Regarding the optimization of these networks, we use the same method as Dreamer [11, Sec. 4] with a caveat: the dynamics is not trained jointly with the environment and the reward. To train the dynamics, we propose two options. In the first option, we do not learn the dynamic model. For instance, one could use an already learned dynamic model and keep it fixed for the whole of the training. This is interesting as the physical model remains constant throughout the training. It helps learning the value and the reward model faster than if we were to learn it from scratch. Also, it offers the possibility to completely remove the learning and neural network components from the dynamic estimation, by using an analytical model, for instance. In robotics, this makes sense, as good analytical/learned models are often already known. If we don't know the dynamics, the second option is to learn it as we discover our environment. However, to ease

the optimization of the actor, the reward, and the value, we do not train the dynamics at the same rate as the rest of our agent's neural networks. We train the dynamics less often, but for more steps than the other components. While this technique does not offer as much stability as the first method, it still increases the overall stability of the learning process.

**Learning behaviors.** With the learning of the system dynamics and environment covered, we can now use them in the imagination process. The imagination generates imagined trajectories from which the actor is learned. In the original Dreamer, imagined trajectories start from latent states $s_t$ obtained from actual observations, and follow the predictions of the transition model $s_\tau \sim q(\cdot|s_{\tau\text{-}1}, a_{\tau\text{-}1})$ and the policy $a_\tau \sim q(\cdot|s_\tau)$. In our case, we use the dynamic transition model as a physic engine for the imagination, and the environment transition model as the rendering engine. This means that to imagine a trajectory, we start from a pair of latent states $(S_\tau^{env}, S_\tau^{dyn})$ obtained from actual observations, and iteratively apply the transitions models given in Eqs. 2 and 3, the policy being given by $a_\tau \sim q(\cdot|S_\tau^{env}, S_\tau^{dyn})$. As this process goes along, the proprioceptive variables $x_\tau$ are reconstructed from the physical state $S_\tau^{dyn}$ using $q_\phi(\hat{x}_t \mid S_t^{dyn})$. They are then used inside the environment transition model. An important note is that, since RSSMs are VAEs, if we sample from the stochastic state to reconstruct the physics, then the imagined proprioceptive variables will be noisy. To prevent that, we take the mode of the distribution which generates smooth physical predictions. Once we have a set of trajectories, we can get each pair of states $(S_\tau^{env}, S_\tau^{dyn})$ and use them to estimate the reward and value associated with them. Using these, we train the policy and the value estimator using the same technique as the original Dreamer [11, Sec. 3].

**Environment Transfer.** Unlike Dreamer, our method allows transferring environments from one robot to another. In this context, we consider that both robots share the same sensors and same state-space. We also consider that the environments in which the robots are going to evolve are similar. Let us consider robot A, with already known dynamics modules $p_\phi^A(S_t^{dyn} \mid S_{t\text{-}1}^{dyn}, a_{t\text{-}1}, x_t)$, $q_\phi^A(S_t^{dyn} \mid S_{t\text{-}1}^{dyn}, a_{t\text{-}1})$. Let us consider robot B, with already known environment modules $p_\eta^B(S_t^{env} \mid S_{t\text{-}1}^{env}, x_{t\text{-}1}, o_t)$, $p_\eta^B(S_t^{env} \mid S_{t\text{-}1}^{env}, x_{t\text{-}1})$, $q_\eta^B(r_t \mid S_t^{env}, S_t^{dyn})$, and a pre-collected set of samples of robot B interacting with its environment. To learn the actor of robot A, using the environment of robot B, we learn entirely offline the actor, the reward, and the value, using the set of experiences collected on robot B. It should be noted that we would not need to retrain the reward if it was rewritten as $q_\eta^B(r_t \mid S_t^{env}, x_t)$. The main disadvantage of this method is that because we are using experiences collected on robot B, the observation of the dynamics will match the dynamics of robot B and not the one of robot A. However, as the imagination progresses the dynamics will match the one of robot A. Another issue can arise if the robots are radically different. In this case, the samples collected on robot B will most likely not explore the state-space of robot A enough to control it perfectly. Nonetheless, as we will demonstrate later, it allows learning good preliminary policies to further refine the agent from. The exact algorithm used to learn the agent policy is given in Appx. C. Besides, the state-space of both robots must be the same, which makes this method mostly aimed towards mobile robots. Examples of the state-space of our robots are given in Appx. D.2.

## 4  Robots, Task & Evaluation

**Robots.** To evaluate our approach we use two robots (Appx. A.1, Appx. A.2): an Unmanned Surface Vehicle (USV) and an Unmanned Ground Vehicle (UGV). The USV is a Clearpath Heron, a small catamaran with a turbine in each hull. There are two action dimensions: the first element controls the left turbine, the second element controls the right one. This means that to drive in a straight line, the agent must send the same value to both turbines. The UGV is a Clearpath Husky, a skid-steer ground robot controlled with a twist input. The size of the action of the Husky is 2, but unlike the Heron, the first element controls the forward velocity, while the second controls the angular velocity. Both robots share similar sensors: a 2D laser-scanner, an RTK GPS and a 9DoF IMU. To prevent damaging our robots' actuators, we saturate their acceleration before applying the output of the policy. More information regarding the exact specifications of the USV and UGV can be found in Appx. A.1 and A.2. We teach our agent to solve a sensor-based navigation task: the robots must follow the shore of a lake at a fixed distance while maintaining a set forward velocity. Hence, the reward generated by the environment is made of two components: a low-level constraint, the velocity, and a high level constraint, the distance from the shore.

**Task.** The exact description of the reward is available in Appx. B. Fig. 3 illustrates the USV solving its task.

To fulfil their tasks, the robots rely on a 2D laser-scanner and on their own velocity, estimated from the fusion of the RTK GPS and the IMU through an Extended Kalman Filter (EKF). Using the EKF, we provide to the agent three proprioceptive variables: its forward-velocity, its lateral-velocity, and its angular-velocity. The action space of both robots is $[-1, 1]^2$.

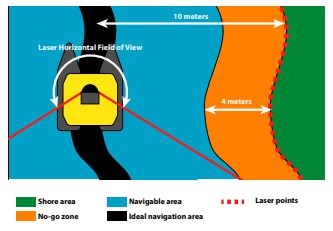

Figure 3: USV/task.

**Evaluation.** We test our method both in simulation and on a real USV. In simulation, to evaluate the performance of our method, we used the different components of the reward as metrics. Each model ran for two hours and at the end of each run we collected the standard deviation, the mean, the 10% quantile and 90% quantile of the rewards. In the real world, we cannot compute the distance reward accurately: our estimation of the distance to the shore is uncertain because of the high level of noise of the laser data in a natural environment. This is particularly true because the experiments were carried out in late spring, where the luxurious vegetation makes laser-scans highly unreliable. For this reason, we will not be evaluating the agent distance from the shore, but we will visually inspect the followed trajectories. The training settings are given in Appx. D

## 5 Simulation Results

We compare our method to the original version of Dreamer, and illustrate the benefits of having a separated state for the dynamics. All models trained with our approach have been trained with a fixed dynamic model. This means that the dynamics was learned ahead of time and could not be refined during the training process. This was done to show that we could have used analytical models instead.

**Unmanned Surface Vehicle.** To show the benefits of using a separated state for the dynamics, we ran a benchmark using the USV. To verify that the agents perform optimally, we first run a test where they are deployed in their original training environment. Then, to evaluate how the agents behave when the dynamics change, we multiply the simulated robots damping factor by two. Increasing this factor makes the robots' dynamics softer: for the same inputs, the acceleration will be slower. To test the robustness to perturbations in the dynamics, we also run a test with the original dynamics and a constant water current with a speed of $0.4m/s$. Finally, to further stress the agents, we multiply the damping factor by two and add $0.4m/s$ of current. The results of this benchmark can be found in Fig. 4.

Our method is more robust than Dreamer to both dynamics change and perturbations. Most notably, it maintains a correct velocity reward across the whole benchmark, when Dreamer fails as soon as the dynamics changes. This suggests that images are not sufficient to approximate the velocity correctly, and that by using our separated physical state, the actor is able to adapt its behavior to match the desired velocity, despite having never seen such events in the training. Moreover, having access to a physical state seems to help to solve the high-level constraints, as our method outperforms the vanilla

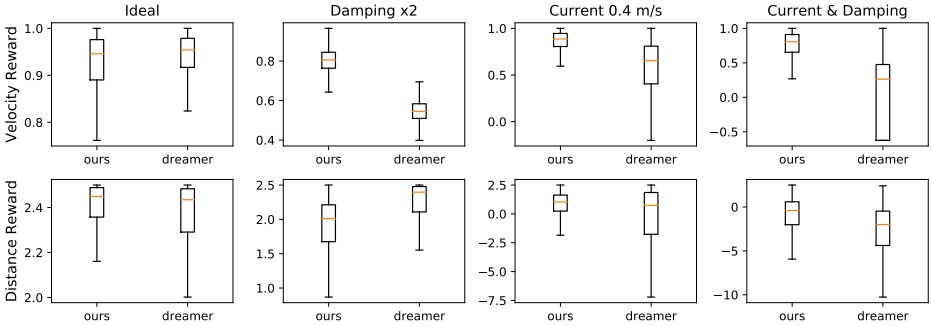

Figure 4: Box plots illustrating the benefits of using a separated state for the dynamics (ours) vs not using one (Dreamer). High value indicates better performances. Small spread indicates better consistency. Results obtained in simulation.

Dreamer on this part of the task too. The only exception is when the damping factor is multiplied by two. In this case, the lower velocity of the robot allows Dreamer to follow the shore better. Overall, these experiments show that splitting the single latent state into two, one for the physics and one for the environment, is not only a viable concept but also increases the robustness of the agent.

**USV to UGV transfer.** To showcase the transfer capacities of our method, we learn an actor completely offline, using the environment of the USV and the dynamics of the UGV. This is possible as we separate the latent-state into those two parts, dynamics and environment. As a reminder, the USV and the UGV do not react to actions in the same way. The USV is driven by sending commands to the left and right turbines, when the UGV is driven by sending forward velocity and angular velocity commands. Furthermore, the dynamics of both systems are radically different: the USV glides, has a huge inertia and has a significant amount of lateral velocity whereas the UGV has almost no slippage, no lateral velocity and no inertia. To train our actor, we first trained our USV to fulfill the shore following task, and we then took its environment model (including the reward) along with the samples of its interaction with the simulation. We secondly learned the dynamics of the UGV from a set of past interactions with the simulation. Finally, using the environment of the USV and the dynamics of the UGV, we trained the actor entirely offline. We then deployed it in simulation and compared it to an agent trained on the UGV and on the environment for an equivalent amount of training steps (300k).

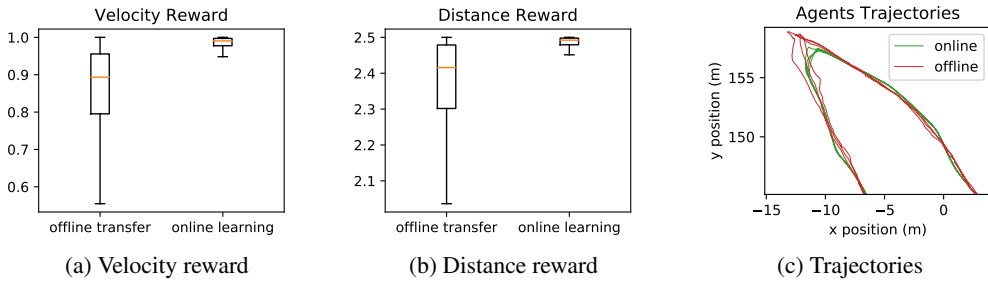

(a) Velocity reward      (b) Distance reward      (c) Trajectories

Figure 5: Environment transfer results. Higher values indicate better performances, smaller spread indicates better consistency. Results obtained in simulation.

Figs. 5a and .5b show the performance of the agent trained offline with environment transfer (offline transfer, ours) compared to the agent trained online with its own environment (Dreamer). It is worth noting that we did not include the results of the USV actor applied directly on the UGV: it completely failed to solve the task because of the complete difference in the commands mapping. From these results, we can see that the agent trained online achieves almost perfect results, with an average distance reward of 2.48. On the other hand, our agent achieves an average reward of 2.4, which amounts to about ±30cm of error on the 10m distance it must keep from the shore. Fig. 5c shows the trajectory of the agents in the hardest spot of the simulation: a narrow hairpin-turn. We can see that the trajectories of the agent learned online are closely packed together. On the contrary, the trajectories of the agent learned offline are more loosely grouped, indicating that this policy is less reliable and most likely less overfitted to the environment. These approximate trajectories can be explained by the fact that the imagination is inaccurate at its beginning. Since the samples used to generate the starting states of the imagination are taken from the USV, these starting states contain dynamics that are not feasible on the UGV. Yet, despite these shortcomings, we demonstrated that we can easily transfer environments in a fully offline fashion even on robots with fundamentally different dynamics and action mappings.

## 6 Real World Results

Finally, we evaluate our method on a real USV. To do so, we train two agents in simulation, one with Dreamer and one with our method, and deploy them directly on the real robot in a zero-shot setup. Thus, the dynamics learned by our agents is only approximately matching the one of the system. We evaluate the performances of both agents on a whole lap around the lake. These laps are shown in Fig. 6a, a lap is about 1.6 km long and takes approximately 20 minutes. During their lap, neither of the agents collided with their environment. Regarding the trajectories, Fig. 6b shows that the trajectory followed by our method is much less winding than the one followed by Dreamer. The reason for

that seems to be that the agent using Dreamer tends to overshoot more than the agent trained using our method. This overshooting behavior can easily be spotted on the cropped image of the lake. On the middle right part of the image, we can clearly see the Dreamer agent overcompensating and navigating toward trees. This behavior cannot be seen on the agent trained using our method, which follows the shoreline at a much more constant distance. It further demonstrates that accessing the dynamics makes the policy naturally robust to changes in the dynamics.

As for the velocity, the right most picture of Fig. 6c shows that the agent with a physical state (ours) is the closest from the desired forward velocity. Additionally, our method shows a smaller spread when compared to the original Dreamer. It is interesting to note that this model can fulfil its task despite having noise on the physical inputs provided by the EKF, which is something it had never encountered before. Furthermore, during the experiments, we observed that our agent was making less brutal accelerations and seemed to have a more fluid way of steering the boat. All in all, this confirms the simulation results shown in Fig. 4: our method is more robust and better matches the forward velocity target.

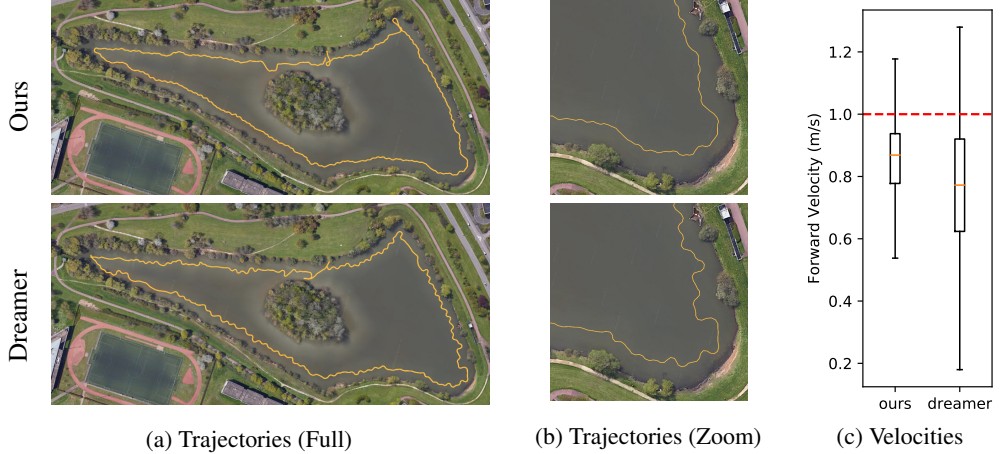

(a) Trajectories (Full)  (b) Trajectories (Zoom)  (c) Velocities

Figure 6: Real world experiment. First row, overhead imagery of the deployment site, with the full trajectory of the agents in yellow. Center row: zoom on the bottom right corner of the lake, the trajectory of the agent can be seen in yellow. Last row, comparison of the forward velocities reached by the two agents. We invite the reader to report to Appx. E for higher resolution images. Overhead imagery from Google Earth, 2021, trajectories plotted using Google Earth KML API.

## 7  Conclusion

In this paper, we showed that splitting the latent-state of MBRL into two sub-latent-states, one for the environment and one for the dynamics, is a viable concept in both the simulated and real world. In all our experiments, our method was trained with a fixed dynamic model which seems to confirm that similar results could be achieved with analytical models, or any other system identification method. Through our experiments, we showed that explicitly adding information about the dynamics of the system make the agent more robust to both perturbations and changes of the dynamics. Finally, we showed that our method could achieve environment swaps from one robot to another, allowing to train robots fully offline. The learned policy could then be deployed on the target system and be refined.

Despite showing strong results, some limitations remain. Most notably, the transfer between systems will only be possible if they share the same state-space. Additionally, with the current setup, the dynamics does not explicitly depend on the environment. This could be addressed by providing either exteroceptive inputs or access to the environment state to the physics module. Future work will focus on integrating this component, along with how to apply domain randomization on the robots' dynamics within the imagination process. This is possible because we can manipulate the dynamics independently of the environment within the imagination while being sure that the imagined environment will be consistent with imagined physics. In doing so, we hope to significantly speed up the learning process of robust actors. We will also investigate the integration of a physical model that uses physics as a model prior as in [26, 22, 27].

**Acknowledgments**

This work was partially funded by the French Agence Nationale de la Recherche (ANR) under the reference number ANR-19-CE10-0011.

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
