# OpenReview forum: "Learning Behaviors through Physics-driven Latent Imagination"
_robot-learning.org/CoRL/2021/Conference — CoRL2021 Oral_

### Official Review · Reviewer_WcEr · 2021-07-11

**Originality:** Good
**Technical Quality:** Good
**Clarity Of Presentation:** Good
**Impact:** 3

**Recommendation:**

Weak Accept: I recommend accepting the paper, but will not argue for my recommendation if the majority of other reviewers have a different opinion.

**Summary:**

The paper proposes a novel algorithm for robot-environment decomposition in MBRL, decomposing the prioprioceptive and exteroceptive modeling. Specifically, the method is based on Dreamer, and learns two RSSM models, one to predict next robot state given action, and another one to predict next visual (or in this case LIDAR) observation given the robot state. It is shown that this decomposition improves robustness to distributional shift, including the sim-to-real shift, with respect to a monolithic Dreamer model. Further, it can be used to adapt to new robots by composing an environment model of target environment with a robot model of a source robot. Experiments on ground and water surface navigation are provided in simulation and real world.


**Issues:**

See comments on Weaknesses. Further, there are the following writing issues:
- l30: Finn’17 should be cited together with [9]
- Decomposition on l41 is quite intriguing. However, as far as I can see it is not used anywhere else in the paper. If the proposed method indeed decomposes the dynamics as on l41, perhaps this can be derived explicitly.
- l196 the reference is broken

Finn’17, Deep visual foresight for planning robot motion


**Reviewer Expertise:**

Very good: Comprehensive knowledge of the area

**Strengths And Weaknesses:**

Strengths:
- The algorithm is novel, sound, and likely will be used by robotics researchers
- The paper is well-written
- The experiments are extensive and clearly demonstrate the benefits of the proposed method

Weaknesses:
- The experiments are perhaps a little unsatisfying. First, the paper does not analyze how exactly the proposed method improves robustness. There are no provided ablations.
- Second, it is unclear why it is necessary to use Dreamer for the proposed use-case. Since the method assumes access to a simulator, a policy can be trained on the simulator rollouts directly using an on-policy algorithm. Similarly, since the simulator is available for all robots, it is easy to just train on the target robot and environment instead of performing transfer.


**Summary Of Recommendation:**

Overall, the strengths of the paper outweigh the weaknesses. The paper proposes a promising algorithm and provides sufficient experiments to validate the benefits of the proposed idea. The algorithm is relevant and is likely to be used by other researchers.

---

> ### Author Response · Authors · 2021-08-21
> **Authors' response**
>
> We thank the reviewer for their review. Please find our responses below:
>
> > Second, it is unclear why it is necessary to use Dreamer for the proposed use-case. Since the method assumes access to a simulator, a policy can be trained on the simulator rollouts directly using an on-policy algorithm.
>
> The main reason we went with Dreamer in the first place was its high performance and its better sample-efficiency. Sample efficiency is important because the simulator that we use, gazebo, is particularly slow: it does not run in real-time with our environment. Another reason is that Dreamer being sample efficient, it would allow us to fine-tune from real-robots with less data than most algorithms. In [2] they provide a comparison of how Dreamer fairs against other popular RL methods (including on-policy methods) on a robotic task. Finally, if one wanted to apply our method on real robots, then learning online is most likely not an option as explained in the response to reviewer Q3uB.
>
> [2] Brunnbauer, Axel, et al. "Model-based versus model-free deep reinforcement learning for autonomous racing cars." arXiv preprint arXiv:2103.04909 (2021)
>
> > Similarly, since the simulator is available for all robots, it is easy to just train on the target robot and environment instead of performing transfer.
>
> We chose to transfer because it alleviates the need to use the simulator, allowing to learn an agent much faster than if we had to interact with the system. Furthermore, the simulator only provides a coarse approximation of the robot dynamics and the environment. Hence, the transfer is important as we can keep the model trained on the simulation environment and fine-tune the agent using a dynamic model learned using data collected on the real-robot. This means that the learning of the task is independent of the learning of the dynamics. Hence, we could learn the dynamics from samples that were collected during past experiments with the robot. Ideally we would have wanted to show that, but we have completely modified our boat’s state estimator for these experiments, which means we had no previous measurement to learn a model from. Another key reason is that in the future we would like to share environment and tasks across a set of robots, and use the same component for the environment while using different dynamic models.
>
> > l30: Finn’17 should be cited together with [9]
>
> Is the reviewer referring to :
> Finn, Chelsea, and Sergey Levine. "Deep visual foresight for planning robot motion." 2017 IEEE International Conference on Robotics and Automation (ICRA). IEEE, 2017.
> If so, it will be added with [9].
>
> > Decomposition on l41 is quite intriguing. However, as far as I can see it is not used anywhere else in the paper. If the proposed method indeed decomposes the dynamics as on l41, perhaps this can be derived explicitly.
>
> We decompose the state as:
>
>  $$S^\text{tot} = (S^\text{env}_t, S^\text{dyn}_t)$$
>
> We also decompose the dynamics as:
>
> $$P(S^\text{tot}_{t+1}\|S^\text{tot}_t, a_t) = P(S^\text{dyn}_\text{t+1} \|S_t^\text{dyn}, a_t) P(S^\text{env}_\text{t+1} \| S^\text{env}_t, S^\text{dyn}_t) $$
>
> We acknowledge that this was not clearly stated in the method, and we will update this section to make the decomposition clearer.

---

> > ### Comment · Reviewer_WcEr · 2021-08-30
> > **Rebuttal Acknowledged**
> >
> > Thanks for the clarifications.

---

### Official Review · Reviewer_Q3uB · 2021-07-24

**Originality:** Good
**Technical Quality:** Good
**Clarity Of Presentation:** Very Good
**Impact:** 4

**Recommendation:**

Strong Accept: I recommend accepting the paper and will argue for my recommendation even if other reviewers hold a different opinion.

**Summary:**

The paper introduces a new extension of a Dreaming, model-based neural network, in which policies are trained by creating synthetic, ie dreamed or imagined, trajectories in a latent state embedding for additional training data.
The contribution of the paper is to split the latent embeddings, and latent dynamics functions into two: One models the environmental transition function while the second models the dynamics of the intrinsic states of an agent.
Both models are then used to unroll two latent trajectories into the future: One predicting changes in the environment, the second changes to the robot or agents own state.
The benefit of this approach is that in principle, either the networks modelling the environmental dynamics or the networks modelling the agent dynamics can be replaced or re-trained. This would allow in principle to speed up the training process of a new agent, or in a new environment, or allow the possibility to transfer to new combinations.
For example, if agent A is trained in environment 1, and agent B in environment 2, one would be able to exchange the learned networks and transfer zero-shot to a combination of, for example, agent A and environment 2.

The paper provides some evidence that this is possible by using the proposed approach to train the environmental and agent dynamics of an autonomous boat and a vehicle. In evaluations, the paper is able to demonstrate the ability to transfer knowledge in a limited experiment in which the environmental dynamics experienced by the boat and the agent dynamics learned from the vehicle.

**Issues:**

- The notation in the introduction, figures and the methods section should be the same.
- The mentioned problem of having different dimensionalities between agents or having additional internal states, such as motor states, should be discussed.
- Ideally, the appendix should give a more detailed explanation of the learning setup, especially regarding the number of states for each task, their dimensionality and how they are defined.
- Appendix D does not exist - this also means that the details reg. the curriculum training process is missing. These details need to be added for the updated version.

**Reviewer Expertise:**

Good: General knowledge of the area

**Strengths And Weaknesses:**

Strengths:
- The general idea of the learning algorithm presented in the paper is interesting and well motivated.
- Although not all presented results indicate that the proposed approach is able to outperform the baseline, ie Dreamer, there is some empirical evidence that the algorithm is working and would be applicable to a range of different scenarios when finetuned (see below)
- The experimental setup is well chosen, and a real-world experiment is conducted in which a real unmanned ship is steered along the shore of a lake.
- The authors promise to publish the code on Github and make it available to the wider research community under an open-source license.


Weaknesses:
- The paper was hard to follow at times. For example, for the dynamics models, the states are first introduced as s^dyn and s^env, later they are changed to greek letters, as well as x and o. Figure 1 uses an alternative notation, with phi, p and h. The notion should be harmonized in the final version.
- The appendix has empty sections.
- The proposed approach works only as intended if the internal robot variables are (a) are of the same type and order, (b) include exclusively state variables that are agnostic to the robot type and model, for example, its world position and velocity, and (c) the state dimensions are the same between different types of robots. The approach works well in the paper because the internal state of the agent contains only the GPS coordinates (position) and task-space velocities and accelerations produced by the onboard IMU. I  believe the proposed approach would have difficulties if the agent state, ie x_t, would also contain actuator states, for example. In such a case, the environmental transition function would also depend on the concrete type of agent - eg in the case of turbines versus wheels. This can be solved in principle by using only a suitable subset of the agent's internal state. This problem should be discussed in the paper to broaden its applicability.
- Environmental and dynamic models are pretrained in simulation. It would have been more interesting if the models were trained online during the learning process.

Others:
- A similar approach in which agent dynamics and environment dynamics are separated from each other, based on graph neural networks, is presented in [1]. This work could be discussed in the related works section

[1] Sanchez-Gonzalez, Alvaro, et al. "Graph networks as learnable physics engines for inference and control." International Conference on Machine Learning. PMLR, 2018.

**Summary Of Recommendation:**

Although the paper has some weaknesses in its writeup, which could be improved, and not all experiments indicate an outstanding performance of the proposed approach, the presented method and its demonstration on a real-world USV should be interesting for the wider robot learning community. The idea of separating the environmental and agent dynamics in a Dreamer-like architecture seems interesting, of value, and well-motivated, especially to allow zero-shot transfer between different agent and environmental combinations.
I am also confident that the authors can address the outstanding issues for their final draft, especially regarding clarity and notation.
It might have been beneficial to apply the proposed method on a few more toy problems in simulation, such as mujoco or pybullet, to demonstrate further its applicability and performance.

---

> ### Author Response · Authors · 2021-08-21
> **Authors' response**
>
> We thank the reviewer for their review. Please find our responses below:
>
> > The paper was hard to follow at times. For example, for the dynamics models, the states are first introduced as s^dyn and s^env, later they are changed to greek letters, as well as x and o. Figure 1 uses an alternative notation, with phi, p and h. The notion should be harmonized in the final version.
>
> We will uniformize the notations.
>
> > The appendix has empty sections.
>
> This is a formatting issue, it will be fixed.
>
> > The proposed approach works only as intended if the internal robot variables are (a) are of the same type and order, (b) include exclusively state variables that are agnostic to the robot type and model, for example, its world position and velocity, and (c) the state dimensions are the same between different types of robots. The approach works well in the paper because the internal state of the agent contains only the GPS coordinates (position) and task-space velocities and accelerations produced by the onboard IMU. I believe the proposed approach would have difficulties if the agent state, ie x_t, would also contain actuator states, for example. In such a case, the environmental transition function would also depend on the concrete type of agent - eg in the case of turbines versus wheels. This can be solved in principle by using only a suitable subset of the agent's internal state. This problem should be discussed in the paper to broaden its applicability.
>
> The reviewer is correct, and we will put the emphasis on the fact that this algorithm was designed with mobile robots in mind. We will add a paragraph regarding limitations in the conclusion.
>
> > Environmental and dynamic models are pretrained in simulation. It would have been more interesting if the models were trained online during the learning process.
>
> We agree with the reviewer, however, with our current robot this is particularly hard to do. Our robot is in the middle of a lake with no Wi-Fi or internet access, and the robot’s battery only lasts one hour under normal use. This means that when training, it would probably drain the battery faster. This makes this process very tedious. We plan to train the dynamic from real samples, but we are still collecting samples. Besides, we think that the zero-shot transfer from simulation to real-world capability of Dreamer is an interesting aspect, even if not a contribution of this paper per se.
>
> > A similar approach in which agent dynamics and environment dynamics are separated from each other, based on graph neural networks, is presented in [1]. This work could be discussed in the related works section. [1] Sanchez-Gonzalez, Alvaro, et al. "Graph networks as learnable physics engines for inference and control." International Conference on Machine Learning. PMLR, 2018.
>
> This is a great suggestion, the paper given in [1] performs system identification using GNNs to model the system and then applies MPC and MBRL to control it. However, we think that this work is different from the one presented here. In [1] the agent performs the control directly on the predictions of the GNNs whereas in our work, we use the output of the RSSM that predicts the physics to drive the RSSM that predicts the environment. We then concatenate the output of their two hidden-states to choose which action to apply. We will add this reference to the related-work to better position ourselves.

---

> > ### Comment · Reviewer_Q3uB · 2021-09-03
> > **Post-Rebuttal Review**
> >
> > I thank the authors for their time and effort in writing their rebuttal. The paper now presents the proposed algorithm fairly and discusses its limitations more adequately, and readability was improved.
> >
> > Similarly to reviewer TKXJ, I think the paper could still be improved by adding more in-depth experiments in simulation over a variety of different tasks. Nevertheless, the real-world experiment conducted by the authors is interesting, and I follow their arguments about the improvement over the classic Dreamer architecture/algorithm. The paper is very suitable for the CoRL conference due to its strong contribution towards robot learning and its focus on real-world experiments with robots in natural environments. It discusses interesting ideas and presents an approach that could inspire further work in this direction.
> >
> > After careful consideration, I will upgrade my rating to a Strong acceptance. I found myself split between a weak and strong accept due to the missing additional evaluations in simulation (as discussed above) for further ablation studies, but ultimately I think the authors did a good job on their real-world experiments and, in conclusion, the positive aspects of the paper outweigh them.

---

### Official Review · Reviewer_TKXJ · 2021-07-24

**Originality:** Fair
**Technical Quality:** Good
**Clarity Of Presentation:** Good
**Impact:** 3

**Recommendation:**

Weak Accept: I recommend accepting the paper, but will not argue for my recommendation if the majority of other reviewers have a different opinion.

**Summary:**

This work performs model based reinforcement learning with a factored latent space, where the environment and robot dynamics are factored into separate components. For the environment component, the forward model uses proprioceptive values instead of agent actions for transitions and representation learning. For the robot dynamics, the robot dynamics ignore environment information for representation and transition values, and is able to reconstruct the proprioceptive values used by the environment model. Model based reinforcement learning follows the pattern of Dreamer, which uses imagined trajectories from rolled out latent states to predict performance.


**Issues:**

Experiments on some domain that stresses a variety of skills would be preferable, such as a robot manipulation domain.

**Reviewer Expertise:**

Very good: Comprehensive knowledge of the area

**Strengths And Weaknesses:**

Strengths:
This work provides a clear, easy to implement system for incorporating robot information without needing too much explicit encoding of information.

The proposal of transfer is very interesting, and the assessment is describes fairly clearly. However, transfer on real robots would have been nice.

The experiments are interesting because they propose robot learning in the middle of a robotics pipeline, (requiring an EKF, noisy GPS), with real world evaluation. This is a much more realistic scenario than many of the other works based on Dreamer have, and demonstration in this way is encouraging.

Weaknesses:

By not including environment information in the robot dynamics, this means that any effect of the environment on the movement of the robot will be encoded as stochasticity (at best). However, the environment can often have a significant effect on the dynamics of the robot. For example, if the robot arm picks up a weight, and this changes the dynamics of the robot, or there is a moving barrier.

Model based reinforcement learning methods, and other reinforcement learning methods, often explicitly encode a separate component of the model for the proprioceptive components of the state. This method is most likely unique in incorporating proprioceptive state into the Dreamer-like framework, but this is a relatively marginal difference from a normal network, hard-coding in robot dynamics, or other methods already being used to include this information.

A component that is often under-appreciated in model based reinforcement learning methods is how to control the distribution of rollout data used to train the model, while the policy might be performing undesirable behavior. This is not particularly well addressed in this work as well, as it describes only a training schedule where the dynamics are updated less. By the nature of the experimental domain, this is probably not stressed, since dynamics in a boat domain do not often significantly change as long as the vehicle does not run aground.

A comparison with a dynamics method that also incorporates the robot information, though in a different way (including it in a larger neural network but a separate channel, including an explicit hand-designed model), would have probably provided a better comparison than Dreamer, since by separating out the robot this work adds additional information that Dreamer does not have, so comparison is limited.

While the boat domain is realistic and interesting, as the only experimental domain (though with simulated versions), it does lack some desirable assessment capabilities. In particular, in this domain the effect of the environment on the agent is minimized, since the effect of the environment on movement of the boat can often be overcome relatively easily. In addition, the dynamics of the agent are intuitive, making it easier to set hyperparameters and balance exploration. It would be desirable to have a robotics domain involving a more complex proprioceptive system.


**Summary Of Recommendation:**

I propose a weak accept, mostly because the work only provides marginal originality, and does not assess on a robotics domain which would have provided more dynamics learning challenge (such as robot manipulation). However, the work provides some interesting results relatively clearly along with the proposed algorithm

---

> ### Author Response · Authors · 2021-08-21
> **Authors' response**
>
> We thank the reviewer for their review. Please find our responses below:
>
> > By not including environment information in the robot dynamics, this means that any effect of the environment on the movement of the robot will be encoded as stochasticity (at best). However, the environment can often have a significant effect on the dynamics of the robot. For example, if the robot arm picks up a weight, and this changes the dynamics of the robot, or there is a moving barrier.
>
> We agree with the reviewer, and we will amend the paper to make sure that the readers are aware of this. More precisely, we will add limitations in the conclusion where we highlight this current shortcoming. We do plan to integrate perception into the physics pipeline when we will be using the husky in a real setup. A possible approach could consist in adding semantic maps and laser as inputs of the physics prediction module.
>
> > Model based reinforcement learning methods, and other reinforcement learning methods, often explicitly encode a separate component of the model for the proprioceptive components of the state. This method is most likely unique in incorporating proprioceptive state into the Dreamer-like framework, but this is a relatively marginal difference from a normal network, hard-coding in robot dynamics, or other methods already being used to include this information.
>
> We do not think our approach to be a marginal modification of existing ones, but we may have missed a part of the literature, and we would appreciate more precise references, and would be happy to better position ourselves in light of this.
> We do think that what makes this method unique is not the incorporation of the proprioceptive state, it is its use. In most MBRL methods, both the proprioceptive information and the environment information are reduced to a single latent space. Here, we use the prediction of the physics component to drive the environment component. This is particularly interesting in Dreamer like frameworks, as we can swap dynamic models to train an agent purely offline without interaction with the simulator as we demonstrate with the husky and heron in simulation.
>
> > A component that is often under-appreciated in model based reinforcement learning methods is how to control the distribution of rollout data used to train the model, while the policy might be performing undesirable behavior. This is not particularly well addressed in this work as well, as it describes only a training schedule where the dynamics are updated less. By the nature of the experimental domain, this is probably not stressed, since dynamics in a boat domain do not often significantly change as long as the vehicle does not run aground.
>
> Adding semantic information regarding the ground in the physics prediction network would probably help address the dynamics changes on robots like the husky. This is something we would like to explore in the future, but beyond the scope of the current paper.
>
> > A comparison with a dynamics method that also incorporates the robot information, though in a different way (including it in a larger neural network but a separate channel, including an explicit hand-designed model), would have probably provided a better comparison than Dreamer, since by separating out the robot this work adds additional information that Dreamer does not have, so comparison is limited.
>
> We agree with the reviewer, and it is possible that if Dreamer is provided with proprioceptive information it may reach performances similar to our version. However, it would not allow transferring the environment or dynamics model. Furthermore, we did not provide a comparison to Dreamer with proprioceptive information as there is no standard method to do so. Depending on how it is implemented the results will be different. With our method,  we show that there are benefits to adding proprioceptive information, and we increase the flexibility of Dreamer like agents.
>
> > While the boat domain is realistic and interesting, as the only experimental domain (though with simulated versions), it does lack some desirable assessment capabilities. In particular, in this domain the effect of the environment on the agent is minimized, since the effect of the environment on movement of the boat can often be overcome relatively easily. In addition, the dynamics of the agent are intuitive, making it easier to set hyperparameters and balance exploration. It would be desirable to have a robotics domain involving a more complex proprioceptive system.
>
> Currently, our research is focused on mobile robots in natural environments, we will make sure to better frame the scope of the algorithm by clearly stating that it is designed for mobile robots.

---

> > ### Comment · Reviewer_TKXJ · 2021-09-08
> > **Response to Authors**
> >
> > The author's responses and addressing of concerns was concise and reasonable. While the limitations of the work remain, the work has been clarified in scope and the differences between Dreamer and other Model based RL methods have been made more apparent. I think this paper adds a clear contribution and interesting experiments.

---

### Official Review · Reviewer_RFKR · 2021-07-25

**Originality:** Good
**Technical Quality:** Good
**Clarity Of Presentation:** Very Good
**Impact:** 3

**Recommendation:**

Weak Accept: I recommend accepting the paper, but will not argue for my recommendation if the majority of other reviewers have a different opinion.

**Summary:**

This paper proposes to separately develop a world model for the robot physics state and the environmental state in the context of model-based RL. The method is particularly developed on the backbone of recurrent state-space model, and case studies robot navigation tasks demonstrate the efficacy of the proposed scheme.

The proposed idea looks interesting and technically sound, and the paper is relatively well-written. However, the contribution of the paper needs to be better clarified, as the baseline idea of separating the physics state and the environment state is not new. Traditional model-based approach actually all rely on this type of separation and it may be the artifact of the effort to try to learn everything in a data-driven way. In this sense, a more fundamental motivation of this work needs to be provided other than the state-of-the-art RL is not working very well.

**Issues:**

- There are some typos.
- Motivation and contributions need to be better clarified.

**Reviewer Expertise:**

Good: General knowledge of the area

**Strengths And Weaknesses:**

Strengths
- The idea is convincing and makes physical senses.
- The paper is relatively well-written and well-organized.

Weaknesses:
- The fundamental motivation is not very clear.
- The contribution needs to be better clarified.

**Summary Of Recommendation:**

This paper presents a technically sound idea that is tested on a reasonable case study.

---

> ### Author Response · Authors · 2021-08-21
> **Authors' response**
>
> We thank the reviewer for their review. Please find our answers below:
>
>  > …, the contribution of the paper needs to be better clarified, as the baseline idea of separating the physics state and the environment state is not new. Traditional model-based approach actually all rely on this type of separation, and it may be the artifact of the effort to try to learn everything in a data-driven way.
>
> To the best of our knowledge, this has not been done in imagination learning. Furthermore, the key difference here is that we use the prediction of the physics to drive the environment. We would appreciate it if the reviewer could provide us any reference we may have missed and we will be happy to add them to the paper so as to better state our contribution.
>
> > The fundamental motivation is not very clear.
>
> We believe the motivation behind this work to be clear, as for example acknowledged by reviewer Q3uB. Our motivation is (lines: 42-46, 57-61, 303-309) to allow for seamless transfer of robots dynamics when they are applied to a similar task and environment. An example could be a mobile robot with a fixed task but with a dynamic model that changes often. Using our method we decouple the learning of the robot’s dynamics from the learning of the environment dynamics. What this means is that these two modules are separated. Hence, if we know the dynamic model of the robot for each payload then we can swap them at runtime, or retrain an agent keeping the environment module without the need for a simulator. This is shown in line (242 - 271).

---

### Author Response · Authors · 2021-08-27
**First Revision**

We thank the reviewers for their comments and suggestions, please find a revised version of the article which addresses the following points:

> By not including environment information in the robot dynamics, this means that any effect of the environment on the movement of the robot will be encoded as stochasticity (at best). However, the environment can often have a significant effect on the dynamics of the robot. For example, if the robot arm picks up a weight, and this changes the dynamics of the robot, or there is a moving barrier. (TKXJ)
> The mentioned problem of having different dimensionalities between agents or having additional internal states, such as motor states, should be discussed. (Q3uB)
 - We have made it clearer that the method proposed here is currently aimed towards mobile robots: l36, l49
 - We amended the method to make it clearer that the physical-states and their dimensions must be consistent between the robots: l175-176 and l189-191
 - We added a discussion about limitations in the conclusion: l309-314.

> The notation in the introduction, figures and the methods section should be the same. (Q3uB)
- These notations have been uniformized: examples can be seen on lines l39-41, l54-55, l115-127, and more

> Appendix D does not exist (Q3uB)
 - This is fixed

> Ideally, the appendix should give a more detailed explanation of the learning setup, especially regarding the number of states for each task, their dimensionality and how they are defined. (Q3uB)
- Appendix D now has a subsection regarding the robots states and their action spaces: l448-455

> The details reg. the curriculum training process are missing (Q3uB)
 - Appendix D now has a subsection regarding the curriculum learning: l459-485

> A similar approach in which agent dynamics and environment dynamics are separated from each other, based on graph neural networks, is presented in [1]. This work could be discussed in the related works section (Q3uB)
- This article has been added to the discussion: l99-l105

> l30: Finn’17 should be cited together with [9] (WcEr)
 - This reference has been added

> Decomposition on l41 is quite intriguing. However, as far as I can see it is not used anywhere else in the paper. If the proposed method indeed decomposes the dynamics as on l41, perhaps this can be derived explicitly. (WcEr)
- We now discuss this in the method l115-120

---

### Meta-Review · Area_Chair_gVmH · 2021-08-14

**Recommendation:** Accept (Oral)
**Confidence:** 4

**Metareview:**

This work claims an idea by decomposing dynamic model into physical agent model and environmental model, which is straightforward but also promising. Authors also demonstrate this kind of decomposition can contribute to transfer efficiently among different agent sharing the same environments. Reviewers have consistent comments that the idea claimed in this work is convincing and also conducted with sufficient experiments overall. However, there are still some concerns raised by reviewer as follows.
1. The contribution and novelty compared to the existing dynamic decomposing work should be more clearly presented.
2. The representation of physical dynamic model sounds like too simple, which would be limited to other more complicated agents.
3.  Experimental setting compared to Dreamer is a little bit unfair.
4. Some notions are not unified.

Although there are still some limitations existed in this work, most of reviewers thought the majority of the contributions have been clearly clarified with consistent opinion to recommend to accept it as CoRL paper.

---

> ### Author Response · Authors · 2021-08-21
> **Authors' response**
>
> We thank the reviewer for their review. Please find our answers below:
>
> > 1. The contribution and novelty compared to the existing dynamic decomposing work should be more clearly presented
>
> We acknowledge that our work may need to be better position itself when compared to other work based on dynamic decomposition. We will add the references provided by the reviewers to better state how our work differs from the rest of the literature.
>
> > 2. The representation of physical dynamic model sounds like too simple, which would be limited to other more complicated agents.
>
> This is correct, this work is a preliminary step to enable more complex dynamic modeling. For now, our work is intended to be used on mobile robots, and we will make it clearer in the introduction. We will also add a paragraph about the limitations in the conclusion.
>
> > 3. Experimental setting compared to Dreamer is a little-bit unfair.
>
> It is true that our model has access to more information than the original version of Dreamer, yet even if the original version of Dreamer had access to this information, it would not be able to perform dynamic model transfer as shown in section V (l242-271).
>
> > 4. Some notions are not unified.
>
> We will unify the notations between the introduction and the method.

---

### Decision · Program_Chairs · 2021-09-13

**Decision:**

Accept (Oral)

**Comment:**

This work claims an idea by decomposing dynamic model into physical agent model and environmental model, which is straightforward but also promising. Authors also demonstrate this kind of decomposition can contribute to transfer efficiently among different agent sharing the same environments. Reviewers have consistent comments that the idea claimed in this work is convincing and also conducted with sufficient experiments overall. However, there are still some concerns raised by reviewer as follows.
1. The contribution and novelty compared to the existing dynamic decomposing work should be more clearly presented.
2. The representation of physical dynamic model sounds like too simple, which would be limited to other more complicated agents.
3.  Experimental setting compared to Dreamer is a little bit unfair.
4. Some notions are not unified.

Although there are still some limitations existed in this work, most of reviewers thought the majority of the contributions have been clearly clarified with consistent opinion to recommend to accept it as CoRL paper.